# Assessing the Impact of Employment in the Informal Sector of the Economy on Labor Market Development

**Anzhelika Viktorovna Karpushkina, Irina Valentinovna Danilova, Svetlana Vladimirovna Voronina * and Irina Petrovna Savelieva**

Higher School of Economics and Management, South Ural State University, 454080 Chelyabinsk, Russia; karpushkinaav@susu.ru (A.V.K.); danilovaiv@susu.ru (I.V.D.); savelevaip@susu.ru (I.P.S.)

\* Correspondence: voroninasv@susu.ru

**Abstract:** The paper analyzes employment in Russia's informal sector based on its spatial and dynamic characteristics. In recent years, the Russian labor market has been characterized by a steady growth of employment rates in the informal sector of the economy, high volatility and territorial differentiation of such employment, and its ambiguous effect on the gross value added and productivity. Given slowing economic growth and reduced productivity, this trend is by no means positive. The database for this study is based on Rosstat data from 83 Russian regions over the period between 2006 and 2020. The research explains the territorial and dynamic features of employment in the informal sector and classifies Russian regions by their employment situation in the informal sector of the economy. We hypothesize that the instability of the labor market is driven by higher rates of employment in the informal sector. To assess employment volatility in the informal sector, we identify the main trends of intersectoral labor mobility and evaluate their intensity. The results distinguish between regions with negative and positive effects. We also reveal problem regions experiencing the negative effects of intersectoral mobility and high or very high rates of employment in the informal sector. The findings can be used to diagnose and monitor regional labor markets, productivity dynamics, and changes in employment as well as to develop national employment programs to ensure the sustainable development of the labor market.

**Keywords:** employment; informal sector; labor market; sustainable development; volatility; reallocation processes; labor productivity; Russia

## 1. Introduction

The growth of employment in the informal sector, territorial differences in its rates, and differences in the cross-industry and the intersectoral mobility of employees between the formal and informal sectors affect the sustainable development of the economy in general and that of the labor market in particular. The Russian labor market is a system of poorly interconnected regional markets. The differences between regional labor markets in Russia include different employment/unemployment rates, job availability, wages, the level of employment in the informal sector, etc. [1]. These values can vary significantly. In 2020, the overall employment rate in Russia was 63.7% (of the total population aged 15 to 72). In the Chukotka Autonomous Area, it was 76.9%, while in Karachay-Cherkessia, it was only 49.0%. The share of people employed in the informal sector may also vary. The total share of employment in the informal sector was estimated as 20.7% (2020), with significant contrasts by region, for example, Moscow, Saint Petersburg, the North Caucasus republics showed employment rates in the informal sector of 4.3%, 7.4%, and 42.2%, respectively.

It is important to study the territorial and dynamic features of employment in the informal sector of regional labor markets as the growth of the informal sector of the economy has an impact on productivity, income, living standards, and tax revenues and leads to the growth of informal employment. The consequences of such growth include

lower capabilities and limited growth drivers of labor productivity as well as lower gross regional and gross domestic products.

It has been reported that the informal sector is a major block to economic development, hindering the country from moving forward and restricting growth and sustainable development [2]. The theoretical aspects of the role of employment in the informal sector are a subject of Russian [3–7] and international research [8–12], and the effects of employment in the informal sector on national and regional economic systems have been thoroughly studied [13–17].

However, the aspects of employment in the informal sector of the economy associated with the heterogeneity of the country's economic space and its impact on labor efficiency and productivity in different regions have not been sufficiently studied. The theoretical problems of the informal sector have been covered the least in countries, such as Russia, that are characterized by a considerable differentiation in the economic space. Regional economic diversity leads to differences in the scale and contribution of the informal sector to the economic performance of the regions.

The Russian economy also exhibits high dynamics that are aggravated by an unstable market mechanism and limited competition between markets. Therefore, we take the impossibility of reducing regional labor force characteristics to the general picture of employment in the national economy and specific the regional characteristics of employment in the informal sector driven by regional socioeconomic conditions into account. This approach allows us to specify and elaborate on different aspects of employment in the informal sector, taking the differences across regional labor markets into account.

Therefore, the purpose of our study is to assess the spatial and dynamic aspects of employment in Russia's informal sector and their impact on economic efficiency and labor productivity in specific regions.

First, we present the theoretical and methodological foundations for our study. Second, we differentiate Russian regions by the level of employment in the informal sector of the economy and determine whether this depends on the local labor market, the sectoral structure of the economy, investment activity, etc. Third, we differentiate Russian regions by the volatility of employment in the informal sector and assess the intersectoral mobility of the employed population and its impact on labor productivity in Russian regions.

The article is structured as follows. Section 2 examines the concept and the existing approaches to the study of employment in the informal sector. Section 3 presents the methods used to estimate the employment rate in the informal sector by region. Section 4 presents the results of the research. Section 5 discusses the findings of the study and offers adjustment solutions for national employment programs.

## 2. Background Theory

### 2.1. Research of Employment in the Informal Sector

We would like to start research on employment in the informal sector with a definition of its scope. Initially, the scope, database, and theoretical findings of the research were formed based on data for the informal sectors of developing Latin American, Asian, and African countries. We mainly paid attention to the differences in infrastructure, the scale of enterprises, and the level of compliance [8]. The concept of the informal sector in the context of employment issues originated in the work of Keith Hart [11], who linked the formation of this sector to low GDP.

The informal sector is a defining issue for developing economics with low levels of per capita production and catch-up development. It is characteristic of underdeveloped regions with an excessive labor force and is bound to disappear with the development of an industrial economy [18].

Having realized that real economic activity exceeds officially recorded levels, developed economies have turned to studying the informal sector. The informal sector forms a substantial part of developed economies (about a third of the official GDP in the US) [19]. However, the reported reasons for the expansion of the informal sector include the 'elusive

nature' of economic activities, where researchers often consider the informal sector to be in the structure of the shadow economy. The 'irregular economy' is an economic activity that saves private costs but compromises public benefits and rights ensured by the laws and regulations governing the relations of property, commercial licensing, employment contracts, relationships of financial credit, and social security [20]. The existence of the informal sector in developed countries is related to the decentralization of production and the pursuance of cheap labor by informal employees to improve business competitiveness [21].

The theory of the informal sector was greatly influenced by the works of Hernando de Soto, which present an original paradigm of the nature of the informal sector based on the theory of excessive bureaucratization and the formalization of business activities that limit their development and distort competitive relations. According to de Soto, such government activities result in an irrational legal regime when the prosperity of an enterprise depends on the costs imposed by law rather than on how well a business operates. The entrepreneur who is better at manipulating these costs or who has ties with officials will be more successful than the one who is only concerned with production [9]. Therefore, the understanding of employment in the informal sector as a niche for low-skilled workers in the Third World has transformed into an alternative method of doing business with its specific features.

According to the International Labor Organization (ILO) [22], the informal sector is a set of unincorporated enterprises owned by households operating in the production sphere and are to be evaluated by a system of national accounts. The informal sector includes unincorporated enterprises without employees and unincorporated enterprises with employees ('enterprises of informal employers'). The ILO defines employment in the informal sector as employment with unincorporated enterprises owned by individuals or households that are not constituted as separate legal entities independent of their owners and for which no complete accounts are available that would permit a financial separation of the production activities of the enterprise from the other activities of its owner(s).

A review of the literature identified the significant characteristics of the informal sector of the economy, which belongs to the household sector. Its distinctive features are (1) the scale of its activity; its non-capital-intensive nature; its size, usually small businesses with a blurred line between household and economic activity and (2) how it differs from the formal sector in terms of compliance with official restrictions in the process of labor activity (registration, conditions, payment, etc.).

In this work, employment in the informal sector of the economy is understood as the involvement of workers in labor and entrepreneurial activity differing in:

(1)  the market nature of the activity;
(2)  the size of the operation—including the number of employees—or self-employment;
(3)  the lack of status as a legal entity.

### 2.2. Research Methods and Approaches of the Employment in the Informal Sector

Following an analysis of the existing research, we identified two major research areas of employment in the informal sector: subject-focused approaches (research on employment in the informal sector) and broader approaches (in particular, research on the shadow economy, which includes the informal sector). The first area studies employment in the informal sector using structural and the institutional approaches. The theoretical foundations of the structural approach in the study of the informal sector and employment include:

(a)  the informal sector being distinguished from the formal sector;
(b)  income of people employed in the informal sector usually being lower than that of employees in the formal sector;
(c)  in most cases, employment in the informal sector offering no social protection;
(d)  the informal sector paying less taxes and does so less often than formal enterprises.

The structural approach uses the concept of the informal sector based on enterprise features rather than individual jobs. In this case, employment in the informal sector refers to employees of unincorporated enterprises and self-employed people.

The institutional approach focuses on the maturity of institutes and their dominance over economic processes; the nature and degree of compliance with formal institutions; and the diversity of institutional interactions in labor relations. In other words, it considers informal employment and looks at the degree to which companies and individuals follow formal rules and labor regulations. An employee is considered to be employed informally if they experience violations of formal, government-imposed institutional constraints related to social security guarantees.

The institutional approach assumes that employees meet the criteria of formal employment in one area and informal employment in another. For example, they pay social contributions but do not use social protection guarantees, or a part of their wage remains undeclared, so no taxes or other charges are levied from that part of the wage.

The second research area includes a large number of publications that study and evaluate the shadow economy (also known as the hidden or non-observed economy), with the informal sector as one of its components. The 1970s and 1980s saw the rise of the monetary approach [19,20,23], which estimates the size of the shadow economy based on financial transactions. An approach based on an analysis of labor market parameters, which is often referred to as the Italian approach (as it is extensively used by the Italian National Institute of Statistics), is used to estimate the scale of employment in the non-observed economy [24]. A common indicator used to estimate the size of the shadow economy and the level of employment in it is electric power consumption [25–27].

An approach focused on the economic and mathematical modeling of the scale of production and the forecasts of structural changes and output can be used to evaluate the size of the informal sector using two groups of variables. The first group estimates the impact on the size and the growth of unmeasured production, while the other identifies undeclared activities. This approach has been used in various studies [28–31] to estimate the size of the shadow economy in New Zealand, Canada, India, and Morocco.

Our review of economic publications summarizes the diversity of methods estimating the employment rate in the informal sector. The use of direct methods, such as dedicated assessment, official observations, surveys, business reports, and statistical reports, evaluate the quantitative parameters of employment in the informal sector. Research is dominated by estimations of the status of and the evaluation of changes in employment in the informal sector as well as the analysis of structural changes in socio-demographic and economic variables [17,32–35].

The assessment of the causes and the size of the informal sector can use indirect estimation methods, such as simulation modeling (MIMIC model), dynamic general equilibrium (DGE), and the statistical economic model [36–39]. For example, [36] identifies the following indicators contributing to changes in the size of the informal economy: taxes/GDP, $M_2$/GDP, and regime durability. The indicators that estimate the scale of the informal economy include the $M_0$/$M_2$ ratio and the growth in electricity consumption. [38] lists several shadow economy growth factors, such as GDP per capita, indirect taxes, unemployment, tax morale, personal income tax, self-employment, and business freedom; the mentioned indicators of the informal economy include GDP per capita, $M_0$/$M_1$ ratio, and labor force participation rate (% of the total working-age population).

Despite the great variety of methods used to estimate the scale of the informal sector and employment in the informal sector, this issue has not been sufficiently covered. For example, there is no comprehensive research on problems of employment in the informal sector that take the territorial features of the economy into account.

### 2.3. Hypothesis

We have generated a hypothesis (Figure 1) that states that territorial characteristics (employment rates in the informal sector, specific regional factors) and dynamic differences (mobility, high employment volatility) can have positive/negative effects on the economy and productivity and can result in better/worse productivity and GRP.

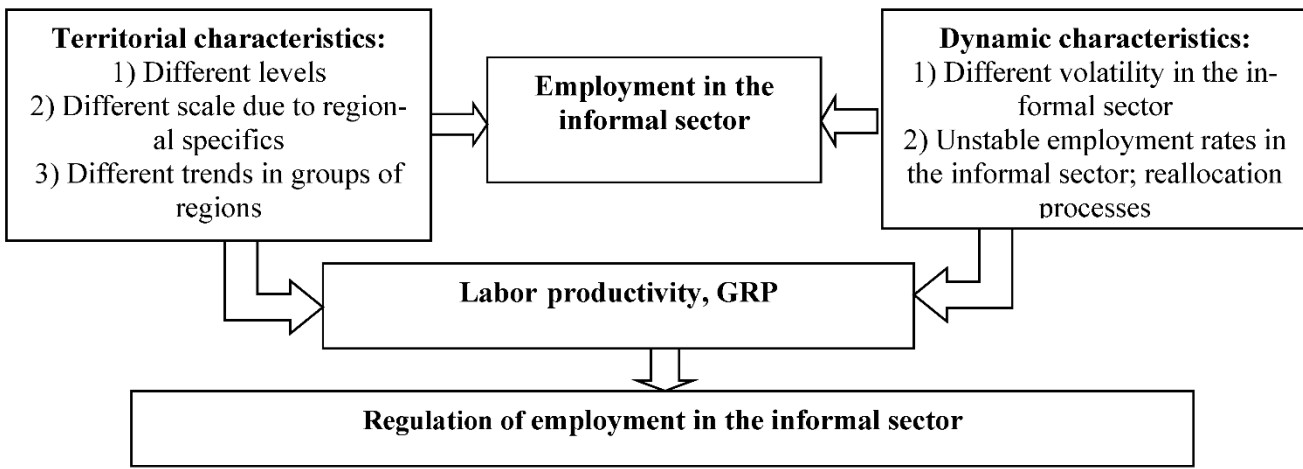

**Figure 1.** Hypothesis.

## 3. Research Methods

Following a review of studies on employment in the informal sector, we concluded that researchers are focused on the level of national economies, including the grouping of indicators, the identification of impact factors, and social and economic consequences. The estimation of employment rates in the informal sector in Russian regions, the identification of factors contributing to employment in the informal sector, and the evaluation of the consequences of the intersectoral mobility of employees has three phases (see Figure 2).

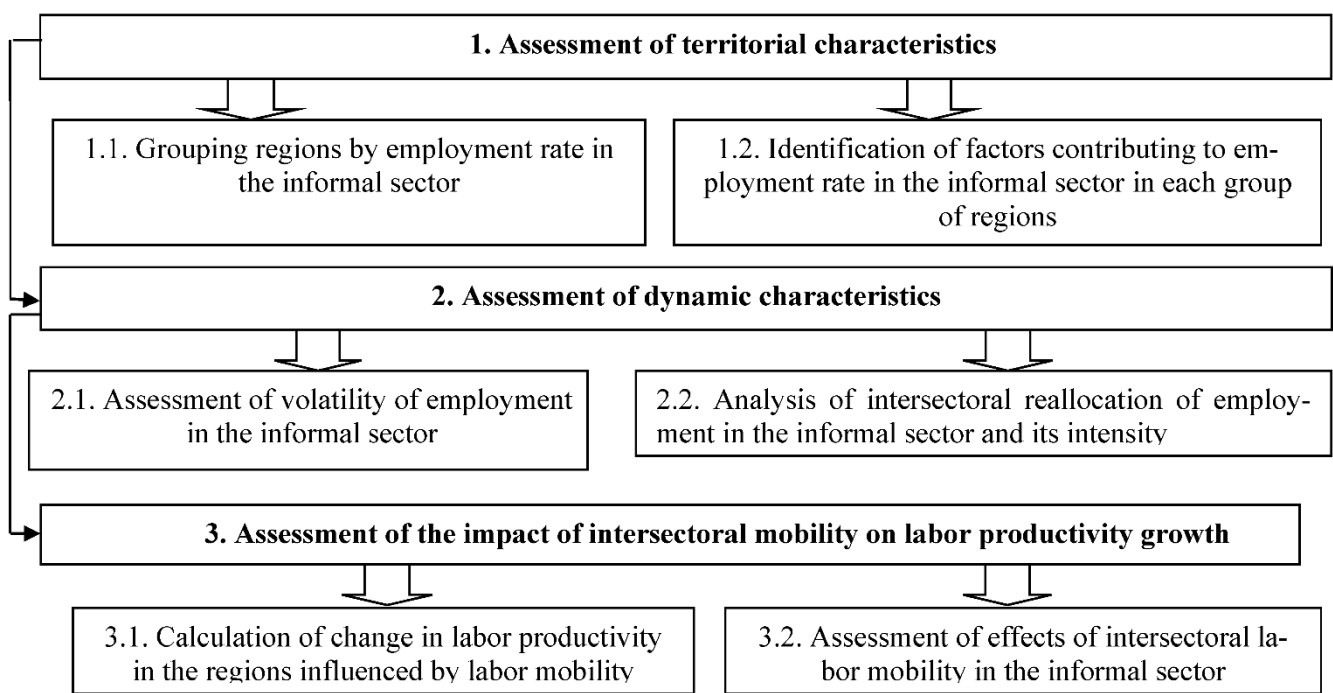

**Figure 2.** Phases of the assessment of employment in the informal sector.

The first phase involves a classification of the regions by the share of employment in the informal sector. This needs to be done due to different levels of employment in the informal sector in different regions.

The classification uses groups with a standard 10% increment; Russian regions can be classified into four groups:

(1)  Regions with low employment rates in the informal sector (up to 10%);

(2)    Regions with average employment rates in the informal sector (10% to 20%);
(3)    Regions with high employment rates in the informal sector (20% to 30%);
(4)    Regions with very high employment rates in the informal sector (over 30% of total labor force).

We identified groups of factors that can affect the scale of employment in the informal sector and tested them using panel data analysis and the model (Formula (1)) for each group of regions.

$$y_{it} = \alpha_i + \beta_1 \cdot X^1_{it} + \beta_2 \cdot X^2_{it} + \dots + \beta_k \cdot X^k_{it} + \varepsilon_{it} \tag{1}$$

where $y_{it}$ is the share of people employed in the informal sector of the economy of the i-th region at time t; k is the number of explanatory variables; $X^k_{it}$ are explanatory variables included in the model for the i-th region at time t (factors); $\beta_k$ are coefficients of the regression equation for the k-th factor; $\alpha_i$ is the permanent member; and $\varepsilon_{it}$ is an error term.

The degree of the impact of each group of factors (aggregated effect) on the resultant figure (employment rate in the informal sector) is the sum of weighted regression coefficients of each factor in the group.

In the second phase, we assessed employment volatility in the informal sector in different Russian regions (Formula (2)) and revealed the types and intensity of labor mobility in the informal sector.

$$\sigma_k = \sqrt{\frac{\sum_{t=1}^{n}\left(E_{is_k} - \overline{E}_{is_k}\right)^2}{n}} \cdot 100\% \tag{2}$$

where $\sigma_k$ is the average square deviation of the number of people employed in the informal sector of the economy of the k-th region; $E_{isk}$ is number of people employed in the informal sector of the economy of the k-th region; $\bar{E}_{isk}$ is the average number of people employed in the informal sector in the k-th region; and n is the number of time periods.

To assess intersectoral labor mobility rates (the mobility of employees between the formal and the informal sectors within a regional labor market; changes in status from 'employed informally' to 'unemployed'; the mobility of employees between the sectors and the transition to unemployed status), we calculated the correlation coefficients between the number of employees in the informal and the formal sectors ($r_{kE_{is}/E_{fs}}$) or the number of the unemployed $r_{kE_{is}/U}$ (Formulas (3) and (4)).

$$r_{kE_{is}/E_{fs}} = \frac{\sum_{t=1}^{n}(E_{fs_k} - \overline{E}_{fs_k}) \cdot (E_{is_k} - \overline{E}_{is_k})}{\sigma_{Efs_k} \cdot \sigma_{\overline{E}is_k}}, \tag{3}$$

$$r_{E_{is}/U} = \frac{\sum_{t=1}^{n}(U_k - \overline{U_k}) \cdot (E_{is_k} - \overline{E}_{is_k})}{\sigma_{U_k} \cdot \sigma_{Eis_k}}, \tag{4}$$

where $r_{kE_{is}/E_{fs}}$, $r_{kE_{is}/U}$ are the correlation coefficients of the k–th region between those employed in the informal sector and those employed in the formal sector or are unemployed, respectively; $E_{isk}$, $E_{fsk}$, and $U_k$ are the number of employees in the k-th region in the informal sector of the economy, employed in the formal sector, and unemployed, respectively; $\bar{E}_{isk}$, $\bar{E}_{fsk}$, and $\bar{U}_k$ are the average number employed in the informal sector, the average number of employed in the formal sector, and the average number of unemployed of k-th region respectively; $\sigma_{Eisk}$, $\sigma_{Efsk}$, and $\sigma_U$ are the average square deviation of the number of people employed in the informal sector, in the formal sector, and the unemployed, respectively; n is the time period.

The intersectoral mobility intensity assessment criterion (using Chaddock's scale [40]) has three levels: high intensity (correlation coefficient over 0.75); average intensity (correlation coefficient 0.5 to 0.75); and low intensity (correlation coefficient under 0.5).

In the third phase, we analyzed the impact of intersectoral labor mobility on changes in labor productivity in the regions. We calculated both the gross value added by employees in the informal sector and the decomposition of changes in labor productivity to assess the effect of intersectoral mobility. The methods for calculating gross value added by employees in the informal sector of the economy of different Russian regions are detailed in [41]. Gross value added by employees in the informal sector ($GVA_{ie}$) is calculated using Formula (5).

$$GVA_{ie} = \alpha_{fce} \cdot GVA^* \tag{5}$$

where $\alpha_{fce}$ is the share of expenditure on the final consumption of households in the k-th region in the actual final consumption of households in Russia and $GVA^*$ is the gross value added of the household sector.

Total labor productivity (LP) can be calculated based on the share of people employed in the formal and the informal sectors using Formula (6).

$$LP = \frac{(GVA_{fs} + GVA_{is})}{E} \cdot \frac{E_{fs}}{E_{fs}} \cdot \frac{E_{is}}{E_{is}} = \frac{GVA_{fs}}{E_{fs}} \cdot \frac{E_{fs}}{E} + \frac{GVA_{is}}{E_{is}} \cdot \frac{E_{is}}{E} = LP_{fs} \cdot \alpha_{fs} + LP_{is} \cdot \alpha_{is} \tag{6}$$

where E and GVA are the number of employees and gross value added in the region as a whole; the values $E_{fs}$ and $E_{is}$ and the values $GVA_{fs}$ and $GVA_{is}$ are the employment rate and gross value added in the formal and informal sectors, respectively; $\alpha_{fs}$ and $\alpha_{is}$ are the share of employment in the formal and informal sectors, respectively.

The decomposition of labor productivity can be calculated based on the share of people employed in the formal and the informal sectors ($\Delta LP$) using Formula (7):

$$\Delta LP = LP^1 - LP^0 = (LP_{fs}^1 \cdot \alpha_{fs}^1 - LP_{fs}^0 \cdot \alpha_{fs}^0) + (LP_{is}^1 \cdot \alpha_{is}^1 - LP_{is}^0 \cdot \alpha_{is}^0) + (LP_{fs}^1 \cdot \alpha_{fs}^0 - LP_{fs}^1 \cdot \alpha_{fs}^0) + (LP_{is}^1 \cdot \alpha_{is}^0 - LP_{is}^1 \cdot \alpha_{is}^0) = (\Delta LP_{fs} \cdot \alpha_{fs}^0 + LP_{fs}^1 \cdot \Delta \alpha_{fs}) + (\Delta LP_{is} \cdot \alpha_{is}^0 + LP_{is}^1 \cdot \Delta \alpha_{is}) \tag{7}$$

where $LP_{fs}^1$, $\alpha_{fs}^1$, $LP_{fs}^0$, $\alpha_{fs}^0$ and $LP_{is}^1$, $\alpha_{is}^1$, $LP_{is}^0$, $\alpha_{is}^0$ are labor productivity and the share of employees in the current (1) and base years (0) in the formal and informal sectors, respectively; and the values $\Delta LP_{fs}$ and $\Delta LP_{is}$ and the values $\Delta \alpha_{fs}$ and $\Delta \alpha_{is}$ are the absolute change in labor productivity and the share of employment in the formal and informal sectors, respectively.

We identified two components of change in labor productivity: changes due to other factors (logistic, organizational, social and economic factors, etc.), ($\Delta LP_{fs} \cdot \alpha_{fs}^0$) and ($\Delta LP_{is} \cdot \alpha_{is}^0$), and changes due to labor mobility between the formal and the informal sectors, ($LP_{fs}^1 \cdot \Delta \alpha_{fs}$) and ($LP_{is}^1 \cdot \Delta \alpha_{is}$).

The growth rate of overall labor productivity ($G_{LP}$) was calculated using Formula (8):

$$G_{LP} = \left( \frac{\Delta LP_{fs} \cdot \alpha_{fs}^0}{LP^0} + \frac{LP_{fs}^1 \cdot \Delta \alpha_{fs}}{LP^0} \right) + \left( \frac{\Delta LP_{is} \cdot \alpha_{is}^0}{LP^0} + \frac{LP_{is}^1 \cdot \Delta \alpha_{is}}{LP^0} \right) \tag{8}$$

where $LP^0$ is the value of aggregate labor productivity in the base year.

To determine the growth rate of aggregate labor productivity, both components of the changes in labor productivity are correlated with the value of aggregate labor productivity in the base year ($LP^0$) in the region (Formula (8)).

Intersectoral labor mobility can have positive and negative effects, i.e., it may lead to a reduction or an increase in labor productivity. The effect sign depends on the total balance of the productivity changes.

## 4. Results

### 4.1. The Assessment of Territorial Characteristics

The assessment of employment rates in the informal sector is based on Rosstat (Federal State Statistics Service https://rosstat.gov.ru/folder/210/document/13204 (accessed on 8 July 2021)) data for 83 Russian regions. We identified four groups of Russian regions by

their employment rates in the informal sector. Figure 3 presents the classification of the regions by the share of people employed in the informal sector.

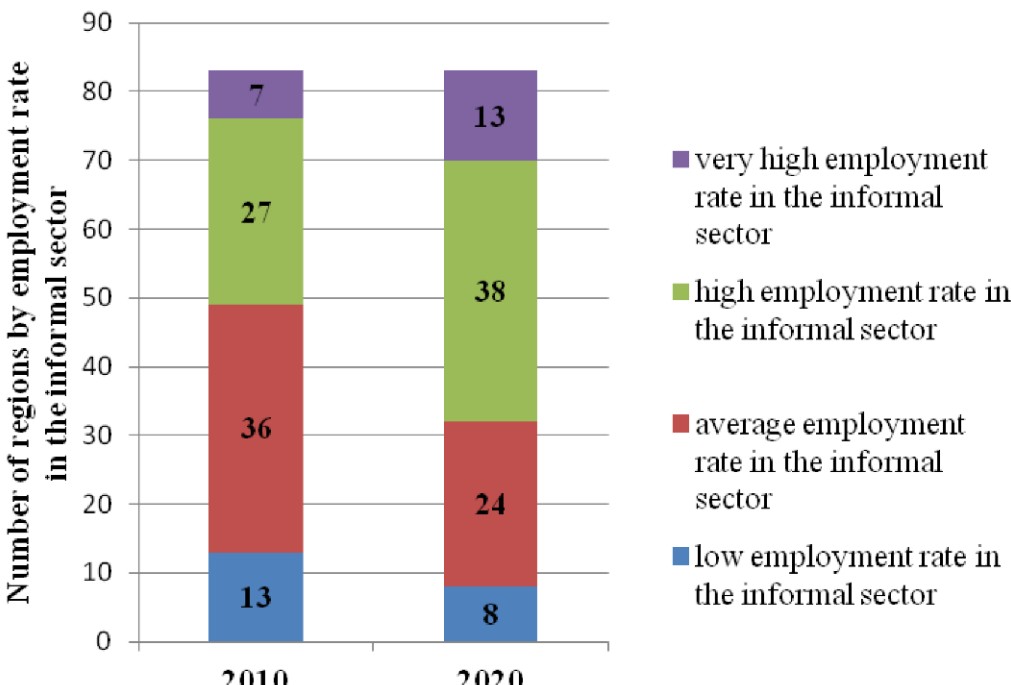

**Figure 3.** Classification of the regions by the share of people employed in the informal sector. Source: Authors' calculations.

There is obvious growth in employment in the informal sector. Between 2010 and 2020, the number of Russian regions with a high or very high level of employment in the informal sector increased from 34 to 49.

Initially, we identified and grouped 39 factors that can affect employment rates in the informal sector based on [14,42,43] and official statistics from the Russian regions. They include 12 labor market factors, 6 sectoral structure factors, 6 investment activity factors, 2 economic efficiency factors, and 4 factors each in institutional environment, migration, household income, and technology.

To test these factors, we built models for each group of regions with different levels of employment in the informal sector. According to the results, the impact of the institutional environment, migration, household income, and technology are insignificant. For the results, see Table 1.

For the results of factor standardization and their impact on employment in the informal sector by groups of regions, see Table 2.

We found that Group 1 (low level) and Group 4 (very high level) exhibit a similar level of impact resulting from the regional labor market factors and the specifics of the sectoral structure of regional economies, while average and high-level groups are dominated by regional labor market factors. These results confirm that regional employment in the informal sector is driven by different factors.

**Table 1.** Modeling results.

| Area of Economy | Parameter | Model 1, Regions with Low Employment in the IS | Model 2, Regions with Average Employment in the IS | Model 3, Regions with High Employment in the IS | Model 4, Regions with Very High Employment in the IS |
|---|---|---|---|---|---|
| | | Coef (Se) | Coef (Se) | Coef (Se) | Coef (Se) |
| Labor market factors | Share of people employed in the formal sector of total economically active age population (X1) | −0.261 *** (0.059) | −0.584 *** (0.040) | −0.696 *** (0.029) | −0.565 *** (0.053) |
| | Unemployment share (X2) | - | −0.460 *** (0.080) | −0.472 *** (0.044) | - |
| | Average monthly wage | - | −0.025 * (0.016) | −0.088 *** (0.035) | - |
| | Employee shortage (X4) | −0.063 *** (0.012) | −0.058 *** (0.02) | −0.016 * (0.012) | - |
| | Dependency ratio (X5) | - | 0.034 ** (0.02) | −0.038 ** (0.017) | 0.128 * (0.083) |
| | Number of graduates with a degree in higher education (X6) | - | 0.028 ** (0.015) | - | - |
| | Number of school graduates (X7) | 0.045 *** (0.012) | - | - | - |
| | Number of college graduates (vocational education) (X8) | - | - | 0.032 * (0.022) | - |
| Factors of sectoral structure | Share of services in the gross value added in the region (X9) | −0.131 ** (0.054) | - | 0.078 *** (0.023) | −0.167 ** (0.084) |
| | Share of trade in the gross value added in the region (X10) | - | - | - | 0.338 ** (0.108) |
| | Share of construction in the gross value added in the region (X11) | −0.152 ** | 0.088 ** (0.043) | 0.078 *** (0.032) | 0.498 *** (0.148) |
| | Share of agriculture in the gross value added in the region (X12) | 0.097 * (0.065) | - | 0.033 ** (0.019) | - |
| | Share of industrial production in the gross value added in the region (X13) | −0.067 ** (0.036) | −0.028 ** (0.012) | - | - |
| Factors of investment activity | Investments in fixed assets (X14) | 0.002 ** (0.0008) | −0.0023 ** (0.001) | −0.004 *** (0.001) | - |
| | Volume of loans granted to sole proprietors (X15) | - | 0.115 *** (0.028) | 0.115 *** (0.026) | - |
| Factors of economic efficiency | Gross regional product per worker (X16) | - | 0.018 *** (0.005) | 0.045 *** (0.01) | - |
| R$^2$ | | 0.841 | 0.683 | 0.724 | 0.838 |
| F-test | | 27.03 | 40.99 | 59.93 | 86.85 |
| F ($p = 0.01$) | | 1.80 | 1.80 | 1.80 | 1.80 |

Source: Authors' calculations. Robust standard errors in parentheses *** $p < 0.01$; ** $p < 0.05$; * $p < 0.1$.

**Table 2.** Aggregate impact of the factor groups on employment rates in the informal sector.

| Impact. | Groups of Regions by Share of People Employed in the Informal Sector | | | |
|---|---|---|---|---|
| | Model 1, Regions with Low Employment in the IS | Model 2, Regions with Average Employment in the IS | Model 3, Regions with High Employment in the IS | Model 4, Regions with Very High Employment in the IS |
| | Aggregate impact of factor groups | | | |
| Factors of sectoral structure | 0.546 | 0.080 | 0.111 | 0.591 |
| Factors of investment activity | 0.002 | 0.081 | 0.070 | 0 |
| Factors of economic efficiency | 0 | 0.013 | 0.026 | 0 |
| Labor market factors | 0.452 | 0.826 | 0.792 | 0.409 |

Source: Authors' calculations.

*4.2. Assessment of Dynamic Characteristics*

To assess the dynamic characteristics of employment in the informal sector, we calculated the volatility coefficient for each region. We then used the arithmetic mean to determine the intragroup employment volatility index for every group of regions (Figure 4).

**Figure 4.** Volatility level by employment rate in the informal sector. Source: Authors' calculations.

Volatility gradually increases from the group with a low level of employment in the informal sector to the group with a very high level. This suggests that the instability of the labor market is driven by higher rates of employment in the informal sector. The main trends of intersectoral labor mobility are:

(1) changes of employment rates in the informal sector related to changes in employment in the formal sector;

(2) changes of employment rates in the informal sector related to changes in unemployment rates;

(3) changes of employment rates in the informal sector related to changes in employment in the formal sector and unemployment rates;

(4) no significant connection between the sectors.

The results of the assessment are presented in Figure 5.

The groups with high or very high employment rates in the informal sector feature high mobility, while the regions with low employment in the informal sector show low mobility. Our hypothesis also assumes that intersectoral labor mobility influences labor productivity.

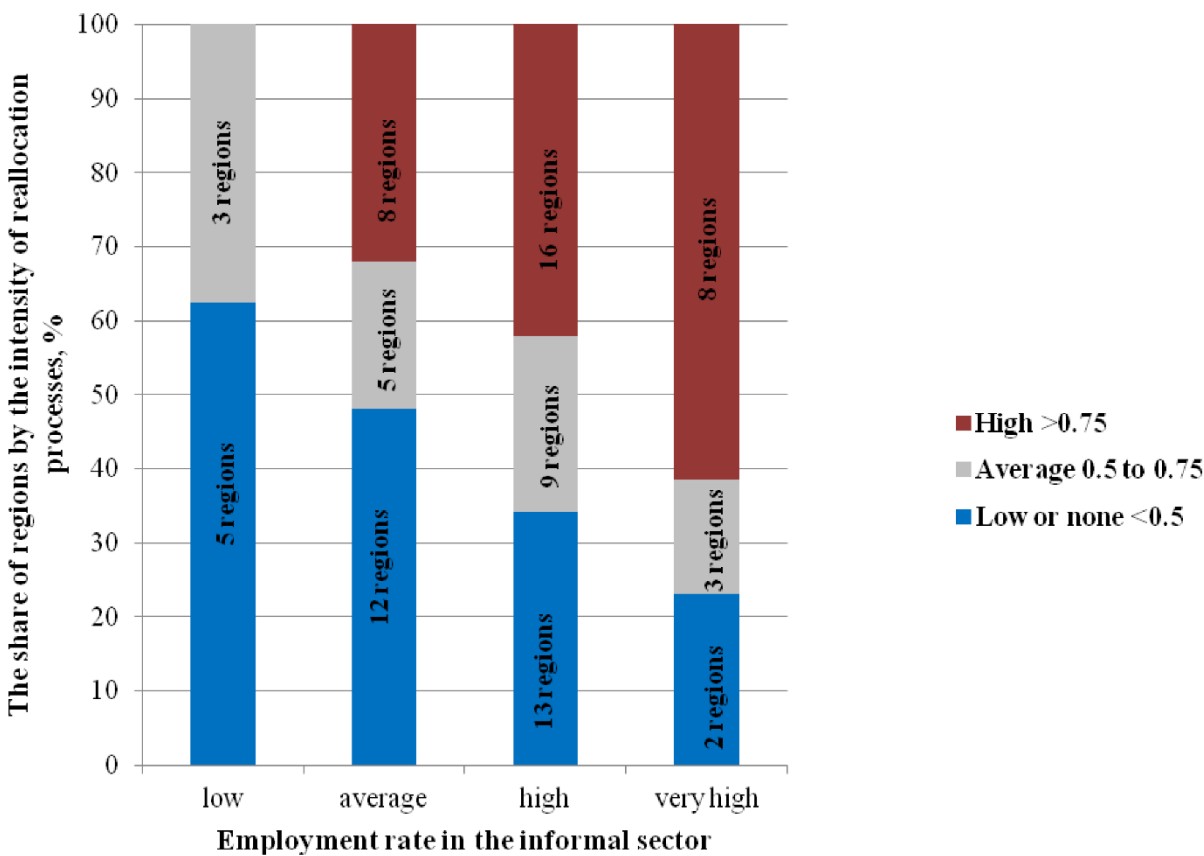

**Figure 5.** Regions by the intensity of reallocation and employment rates in the informal sector. Source: Authors' calculations.

*4.3. Assessment of the Impact of Intersectoral Mobility on Labor Productivity*

An extract from our calculations within the third phase, demonstrating the impact of intersectoral labor mobility on labor productivity, is presented in Table 3.

**Table 3.** Factors contributing to aggregate labor productivity in the Russian regions (extract).

| Regions | Growth Rate of Aggregate Labor Productivity ($G_{LP}$, %) | FACTORS of Changes in Overall Labor Productivity | | | | $r_{kE_{is}/E_{fs}}$ |
|---|---|---|---|---|---|---|
| | | $\Delta LP_{fs} \cdot \alpha_{fs}^{0}$ | $LP_{fs}^{1} \cdot \Delta\alpha_{fs}$ | $\Delta LP_{is} \cdot \alpha_{is}^{0}$ | $LP_{is}^{1} \cdot \Delta\alpha_{is}$ | |
| **Group A: Recipient regions with a negative reallocation value** | | | | | | |
| Astrakhan region | 394.240 | 345.317 | −7.513 | 52.658 | 3.783 | −0.84 |
| Nenets Autonomous Okrug | 365.150 | 374.249 | −16.370 | 4.048 | 3.091 | −0.68 |
| **Altai Republic** | 274.110 | 266.145 | −64.232 | 36.482 | 35.723 | −0.96 |
| **Republic of Ingushetia** | 56.240 | 77.212 | −33.967 | 3.173 | 9.757 | 0.87 |
| Yaroslavl region | 265.690 | 216.623 | −0.869 | 49.134 | 0.802 | −0.50 |
| **Group B: Donor regions with a positive reallocation value** | | | | | | |
| Amur region | 183.920 | 115.232 | 14.086 | 68.459 | −13.857 | −0.81 |
| Volgograd region | 217.880 | 154.059 | 16.963 | 60.973 | −14.117 | −0.50 |
| Penza region | 327.300 | 224.923 | 28.197 | 91.549 | −17.373 | −0.92 |
| Tyumen region | 116.930 | 75.142 | 19.132 | 30.954 | −8.294 | −0.83 |
| Chuvash Republic | 202.730 | 123.457 | 27.263 | 75.718 | −23.708 | −0.66 |

Source: Authors' calculations.

Altai Republic has an overall productivity growth rate of 274.11%. This figure includes productivity growth of 266.14% in the formal sector and 36.48% in the informal sector due to logistic, organizational, social, economic, and other factors and also shows a productivity decrease of 64.23% in the formal sector and a growth of 35.72% in the informal sector due to the transition of employees from the formal to the informal sector. Thus, without the reallocation of employees from the formal sector to the informal sector, which contributed to the final loss of productivity, the aggregate labor productivity growth rate in the Altai Republic would have been 28.5% higher ($-64.2\%$ + 35.7%). The aggregate labor productivity growth rate in the Republic of Ingushetia also declined by 24.21% ($-33.97\%$ + 9.76%) due to intersectoral mobility.

This analysis of intersectoral labor mobility demonstrates both positive and negative effects of reallocation on overall productivity. We identified 28 regions with predominantly negative effects of mobility associated with labor reallocation from the more productive formal sector to the less productive informal sector; we also found 16 regions experiencing predominantly positive effects. These regions are experiencing a reallocation of employees from the informal to the formal sector and, hence, are experiencing productivity growth.

We also calculated the ratio of gross value added in the regions with positive and negative mobility effects in different Russian regions. The regions with a negative effect create two thirds of the total gross value added, which confirms our hypothesis regarding the impact of reallocation on the efficiency of the Russian economy.

## 5. Discussion and Findings

Employment in the informal sector is a separate labor market segment that has a significant impact on both employment rates and the social and economic environment in Russia, in general, but in the regions in particular. Our interest in employment in the informal sector is explained by its dual nature: it is an alternative to unemployment while often being accompanied by informal labor relations, tax evasion, lack of social protection, low-tech jobs, and, as a result, low productivity.

This research identifies the differences in employment in the informal sector between different Russian regions. It confirms that countries with high territorial differentiation exhibit significant regional differences in employment in the informal sector [38,44,45].

Our assessment of employment in the informal sector in Russia leads us to the following conclusions:

Regions with low rates of employment in the informal sector are dominated by the impact of labor market factors and the sectoral structure factors. This group includes regions with a large number of jobs in the formal sector concentrated mainly in the mining industry and the financial service sector. Such regions are characterized by low employment volatility in the informal sector and a low transition of employees from the formal to the informal sector. All regions within this group demonstrate negative mobility.

The informal sector in the regions with average rates of employment in the informal sector is mainly driven by labor market factors. Such regions are dominated by manufacturing, which mainly provides formal jobs. With an average employment volatility rate in the informal sector, they are distinguished by a low reallocation intensity. Most regions feature negative mobility.

The size of the informal labor market in the regions with high rates of employment in the informal sector is also mostly driven by labor market factors. Such regions are dominated by manufacturing as formal jobs; however, they are also characterized by a high share of the service sector, where many informal jobs are concentrated, in the gross regional product. Employment volatility in the informal sector of this group is average with high intersectoral labor mobility. These regions feature both positive and negative mobility effects to a similar extent.

Regions with very high rates of employment in the informal sector are affected by labor market and sectoral structure factors, the latter affecting them slighting more. These regions feature the lowest rates of gross regional product per capita and the highest unemployment

rates. The economic activities with the largest share of gross value added in such regions include wholesale and retail trade, construction, and agriculture, where most jobs are concentrated in the informal sector. Such regions are characterized by high employment volatility in the informal sector and usually high reallocation intensity, i.e., transition of employees from the formal to the informal sector.

Employment in the informal sector reduces unemployment rates and can be considered as a tool to support employment in the economy in general. However, employment in the informal sector is related to lower tax and insurance proceeds to the national and regional budgets, lower productivity and, as a result, limited opportunities for economic growth.

The regulation of employment in the informal sector involves the legalization of employment [46], the improvement of efficiency, and the reduction of unemployment and institutional obstacles to the development of small businesses and self-employment. Another effective tool is a flexible tax policy that can have a rapid and significant impact on employment rates in the informal sector. A higher tax burden can cause legal entities to go underground [38].

This research of employment in the informal sector in Russian regions has revealed problem regions that require the regulation of employment in the informal sector. This includes regions with high or very high levels of employment in the informal sector, where growth in employment in the informal sector leads to a reduction in aggregate productivity.

Therefore, we propose amending the Strategies for Socio-Economic Development of Russian Regions and the National Employment Promotion Program, and in particular, propose the revising of these measures to help neutralize the negative consequences and destabilizing effects of employment in the informal sector.

In our opinion, the scope of regulation and the measures that can help neutralize the potential adverse economic and social effects include the support of entrepreneurship, fiscal regulation, and institutional regulation (the development and enforcement of labor laws, the improvement of employment services, and the implementation of employment programs).

## 6. Conclusions

Employment in the informal sector is caused by a variety of factors and has varying impacts. For countries, including Russia, with a differentiated economic space, it is important to promote research on the geographic differences and the scale and volatility of employment in the informal sector across regional markets. In addition, ongoing research on the impact of employment in the informal sector on labor productivity at the macro level [13] has stimulated similar studies at the local level, which can further be categorized into regional contexts. Testing our hypothesis, i.e., that spatial characteristics and dynamic differences in employment in the informal sectors of Russian regions generate different impacts (positive/negative) on labor productivity, allowed us to make the following conclusions:

Formal employment begins to dominate in the labor market as a region grows and develops, while employment in the informal sector of the economy goes down significantly. Similar findings were obtained in [47]. Our study also confirmed the relationship between the scale of employment in the informal sector of the economy and regional differences in the sectoral structure of the economy, investment activity, supply and demand in the labor market, institutional conditions, and the intensity of the intersectoral flows of the employed workforce. We found confirmation of our hypothesis that the flow of the workforce from the formal sector to the informal sector impacts labor productivity. Similar results were presented in [48]. We also revealed the prevalence of regions experiencing a predominantly negative impact of workforce flows (caused by the redistribution of labor from the more productive formal sector to the less productive informal sector) over the regions experiencinga predominantly positive impact (where there is a transition of the employed from the informal sector to the formal sector and growing labor productivity).

Regions with a negative impact of reallocation on labor productivity must intensify their regulation of the labor market and identify ways to mitigate this trend.

Given that employment in Russia's informal sector is significant and continues to grow, it is important to study its causes and impacts on overall economic growth. Therefore, our study contributes to building a foundation for an in-depth understanding of the problem of employment in the informal sector of the economy and its impacts on the sustainable development of the labor market and the economic growth of Russian regions.

This area needs further research in terms of the ongoing monitoring of the rates of employment in the informal sector in Russian regions to ensure the timely neutralization of potential adverse effects related to reduced productivity in those regions. This area is important for the sustainable development of the labor market in Russia and some former Soviet countries.

**Author Contributions:** Methodology: A.V.K., I.V.D. and S.V.V.; validation, S.V.V., A.V.K.; formal analysis, I.V.D.; investigation, A.V.K. and S.V.V.; resources, I.V.D. and I.P.S.; data curation, S.V.V.; writing—original draft, A.V.K. and S.V.V.; writing—review and editing, A.V.K. and I.V.D.; visualization, I.P.S.; supervision, I.V.D.; project administration, I.P.S. All authors have greatly contributed to the completion of the manuscript by conceiving and designing the research, writing, and improving the article. All authors have read and agreed to the published version of the manuscript.

**Funding:** This research received no external funding.

**Institutional Review Board Statement:** Not applicable.

**Informed Consent Statement:** Not applicable.

**Data Availability Statement:** Publicly available datasets were analyzed in this study. This data can be found here: https://rosstat.gov.ru/.

**Acknowledgments:** The authors thank the editor and reviewers for their informative comments. The authors thank the South Ural State University (SUSU) for its support.

**Conflicts of Interest:** The authors declare that there are no conflicts of interests.

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
