# Peer review of "Assessing the Impact of Employment in the Informal Sector of the Economy on Labor Market Development"

_sustainability, doi:10.3390/su13158435_

Round 1

Reviewer 1 Report

The article deals with a topic of perennial interest in economic research, namely the assessment of the impact of employment in the informal sector or the gray economy. The paper is divided into five sections: introduction, background theory, research methodology, results, discussions and findings. An inclusion of a separate section with conclusions would be welcome! Table 1 is the only one that shows the source. A presentation of the source for the other figures and tables would strengthen the readers' belief that the authors assume their content.

Author Response

Thank you for your valuable comments. We will be sure to cite the source in all figures and tables. All data presented in the tables and figures were calculated by the authors (Source Author's calculations).  We have highlighted and supplemented the conclusions in a separate section. All this is reflected in the corrected version of the article.

Reviewer 2 Report

Dear authors,

I appreciate having the opportunity to review the manuscript entitled “Assessing the Impact of Employment in the Informal Sector of the Economy on Labor Market Development” (sustainability-1253712).

Although the authors have made considerable efforts to develop this paper, however, I believe that the current version of manuscript should be improved through significant revision and re-writing. I want to provide some suggestions for the improvement of this paper as follows.

[1] Introduction

- I think that the overall structure and writing of introduction part are not clear and well-aligned because it is not easy to catch what the research questions and strategies to deal with those are. Please clearly describe those things. As you already knew, the introduction section is one of the most important parts to not only draw attentions of readers but also provide guidelines for them to facilitate a clear understanding of the paper.

[2] Theories and hypotheses

- This paper did not provide the part of “Theory and Hypotheses. So, it is very difficult for me to be sure that the research has an enough level of theoretical value and contribution. I think that this is the critical flaw of this paper. Please provide the part in an elaborated way.

- Although this paper dealt with interesting phenomena, it did not provide adequate theoretical background and support for the development of its hypotheses. This is the critical limitation of this paper. Please clearly explain what its hypotheses are.

[3] Strengths and Limitations of the Study

- Although the authors have attempted to explain the contributions and implications of the paper, I think that the overall quality of the explanations is low. Please provide more elaborated explanations to demonstrate its theoretical and practical contributions.

 I wish these comment may help you to improve your paper. Good luck.

Reviewer 3 Report

When reviewing scientific papers for publication, I usually start with a general overview in terms of a structure, abstract, literature review, methodology, findings of the research, discussion, conclusions, as well as limitations of the study and future directions of the research. I also pay attention to the language level, especially if the paper is written in English, and English is not the native language.

The topic (Assessing the Impact of Employment in the Informal Sector of the Economy on Labor Market Development) can be considered as actual and valuable from the point of view of further research in the area.

There are my reservations to the manuscript.

  1. The reviewed paper entitled “Assessing the Impact of Employment in the Informal Sector of the Economy on Labor Market Development” is generally structured in a proper way. There are, however no sections ‘Conclusions’  with contribution to the theory , future directions of the research and limitation of the study. These sections should be added too, given this is a research paper.
  2. It should be done acc. to the 'from general to details' rule, so first 1-2 introductory sentences, then the purpose of the paper, methodology and finally main findings. There are some introductory sentences and general information about the topic being investigated. In turn, there is neither presentation of the research sample. Key findings of the research are presented, however up to a point only. You should give more details on it.
  3. Introduction is underdeveloped. Where is the research gap and its justification?
  4. The main problem with the paper is Discussion section. This section should discuss the results achieved; In addition, there should be references to the results of other scholars. Unfortunately there is almost nothing in this part, and the second aspect is missing at all.
  5. The literature review is quite good. Generally I claim that Author (s) provide solid theoretical foundations for the analysis using appropriate references. I would, however, recommend to add some references devoted to the  latest literature associated with the topic in question (including Web of Science and Scopus papers).
  6. I also recommend language proofreading by a native speaker.

Round 2

Reviewer 2 Report

Dear authors,

 Thank you for your efforts to improve you paper. However, I am sorry to

tell you that I could not find the theoretical improvement as well as the

adequate level of revision in the discussion part in your revised manuscript. 

Reviewer 3 Report

The main problem with the paper is Discussion section. This section should discuss the results achieved; In addition, there should be references to the results of other scholars.